# Assessment of the Destruction of a Fibre Cement Board Subjected to Fire in a Large-Scale Study

**DOI:** 10.3390/ma15082929

**Published:** 2022-04-17

**Authors:** Krzysztof Schabowicz, Paweł Sulik, Tomasz Gorzelańczyk, Łukasz Zawiślak

**Affiliations:** 1Faculty of Civil Engineering, Wrocław University of Science and Technology, Wybrzeże Wyspiańskiego 27, 50-370 Wrocław, Poland; krzysztof.schabowicz@pwr.edu.pl (K.S.); tomasz.gorzelanczyk@pwr.edu.pl (T.G.); 2Instytut Techniki Budowlanej, Filtrowa 1, 00-611 Warszawa, Poland; p.sulik@itb.pl

**Keywords:** ventilated facades, large-scale model, fibre cement boards, fire exposure, acoustic emission method

## Abstract

This article presents the results of a study involving the assessment of the structural destruction of fibre cement boards taken from a large-scale model subjected to fire. These were subjected to a three-point bending test using the acoustic emission method. The analysis of the obtained results took into account the course of bending stresses σ_m_, the modulus of rupture (MOR), the number of acoustic emission events N_zd_ and the sum of EA events ∑N_zd_. The conducted tests showed that the samples subjected to fire exhibited a clear decrease, up to 72%, in the recorded sum of EA events compared to a reference board (not subjected to fire). The analysis of the obtained modulus of rupture (MOR) values showed a similar trend—the reduction of the modulus of rupture for the degraded samples was in the range of 66–96%. In order to illustrate the changes taking place in the structure of the tested plates more precisely, analyses were carried out using the optical method and a digital microscope. This method may be sufficient for the final evaluation of degraded samples.

## 1. Introduction

Fibre cement boards are products used in construction since the early 20th century. In the 1990s, they underwent a transformation, whereby asbestos fibres, which constitute a health hazard, were replaced by other fibres, mainly cellulose fibres. The fibre cement boards manufactured nowadays consist of cement, cellulose fibres, synthetic fibres, and various additives and admixtures, i.e., limestone dust, mica, perlite, kaolin, microsphere [1,2,3,4]. Fibre cement boards are classified as composite construction materials and are defined as any multiphase material that exhibits a significant portion of the properties of both constituent phases and has been artificially made. The individual constituent phases must be chemically distinct and separated by a distinct interface [5]. Composite materials consist of two phases: the first is the matrix phase, which is continuous and surrounds the second phase, called the dispersed phase (reinforcing elements).

Composite materials are divided into particle-reinforced, fibre-reinforced and structural composites. Fibre cement boards are classified as fibre-reinforced composites, and the phases are dispersed in each direction. The classification of composites is shown in Figure 1.

The main goal in the design of fibre-reinforced composites is increased strength and/or rigidity without increased weight. Fibre-reinforced composites with exceptionally high strengths often contain low-density fibres. The most important factor that increases their strength is the appropriate fibre length (not too short). The mechanical and strength properties of these composites depend not only on fibre properties, but also on the extent to which load is transferred to the fibres by the matrix. The manner of that transfer is shown in Figure 2.

As fibre-reinforced composite materials, fibre cement boards have a matrix phase; in their case, the matrix is based on Portland cement which is responsible for binding the matrix and giving it durability. The second phase is the dispersed phase in the form of fibres. Apart from Portland cement, the matrix contains additional components and fillers, i.e., limestone dust, mica, perlite, kaolin, microspheres. The dispersed phase is characterised by the discontinuous and randomly oriented distribution of fibres. Fibres used in the manufacture of these composites include cellulose fibres, PVA (poly(vinyl alcohol)) synthetic fibres and PP (polypropylene) fibres. Most fibre cement boards make use of all these fibres, with each fibre serving a slightly different role. Cellulose fibres form a spatial mesh that reinforces the entire composite, polyvinyl alcohol (PVA) fibres are added to increase the strength and durability of fibre cement boards, especially those intended for outdoor use.

Fibre cement composites are used both indoors and outdoors. Outdoors, they are used mainly as façade cladding. They are expected to comply with a number of performance requirements referred to in harmonised standards [6]. These boards must also meet a number of resistance and strength requirements under fire conditions. As shown in [7,8], the strength of fibre cement boards after façade fires is significantly reduced, often preventing the ensuing safe use of such a façade. For cladding exposed to fire, it would be necessary to define a zone that requires the complete removal of the façade, including the partial disposal and partial use in other construction elements that do not have to meet such drastic requirements.

## 2. Literature Review

High temperatures have an extremely degrading effect on most building materials including composites (such as fibre cement boards). Composites characterised by two phases in view of the preliminary analysis and identification of degradation after high-temperature exposure can be analysed as separate, independent phases: the matrix phase and the dispersed phase. After this preliminary identification, the composite should be analysed and tested as an integrated, homogeneous product.

In terms of the cement matrix, the analysis of post-fire degradation can be based on studies of concrete samples that are used in many building structural elements exposed to fires. The strain analyses carried out in works on degraded concrete [9,10,11] indicate that temperatures in the range of 300–400 °C are not critically degrading to concrete as opposed to boards, due to the presence in concrete of coarse aggregate fractions, and it is important that the samples are volumetric elements (three of the same order of magnitude). The authors subjected several different cement variants to high-temperature annealing tests in a study [10]. CEM I 42.5 R, a fast-setting and fast-maturing variant of Portland cement, showed poor performance in terms of resistance at high temperatures. The cement matrix in the experimental tests showed residual strength at 500 °C [10]–600 °C [12], while at 800 °C [10], the samples disintegrated spontaneously. Other studies [10,12] demonstrated that any temperature higher than 400 °C [12] has a negative effect on the strength of the matrix. The regression in cement strength starts almost immediately; initially, however, these values are relatively small. Thus, it can be assumed that significant strength regression begins at temperatures above 100 °C [10,13]. It is worth noting that the strength of concrete in a fire situation is also affected by the type of aggregate used.

In the dispersed phase, cellulose fibres, PVA synthetic fibres and PP fibres are used in fibre cement boards. The individual fibres have the following melting points: synthetic fibres PVA (polyvinyl alcohol)—about 200–220 °C [14,15]; PP (polypropylene) fibres—about 175 °C [14,16]; cellulose fibres—260–270 °C [17]. In the case of fibre cement boards, there is a lack of knowledge about the behaviour of this composite and how it degrades at fire temperatures. Most of the scientific literature analysing fibre cement boards focuses on the production process, its possible subsequent optimisation in terms of fibres and the testing of its basic physical properties. One of the few examples is the study presented in [18], where it has been shown that fibres in fibre cement boards degrade at 230 °C only after approx. 3 h of exposure. Damage to such degraded boards, during a three-point bending test, occurs through high-energy brittle fracture. It is noticeable that for fibre-reinforced cement composites, the modulus of rupture rises as the temperature increases up to about 300 °C [19] in a short period of time. It is, therefore, reasonable to believe that such temperatures are safe for these boards over the short term. Destruction of the fibres and cement matrix at a temperature of around 300 °C only occurs after a long period of time. This is influenced by the protection (encapsulation) of the fibres by the matrix phase (cement or concrete). These temperatures correspond approximately to the melting point of cellulose fibres. Szymków carried out tests on fibre cement board samples at 400 °C [18]. The samples showed much less stability at this temperature and degraded at a faster rate. The results exhibited large discrepancies because, depending on the manufacturer, the ingredients and the manufacturing technology, some samples “lasted” a maximum of several minutes while, while others were destroyed during the test.

Based on a study of large-scale models, Schabowicz et al. [7] showed the percentage of loss of strength parameters of fibre cement boards under the influence of fire. These studies demonstrated differences from model studies on small samples, but the temperatures at which degradation started, i.e., around 200 °C, was consistent for both types of study. Large-scale studies, compared to studies on small-scale samples, established different times for the occurrence of significant degradation. This is mainly due to the fact that studies on small-scale samples are mostly characterised by constant and uniform temperature effects. Hence, it follows that the structural expansion of fibre cement boards at high temperatures and the duration of action on such products is critical.

Failure to maintain the production regime (e.g., maintaining the appropriate temperature and humidity) may lead to fluctuations in the strength of concrete during a compression test, even up to 20%, which may cause significant differences in reference samples [20,21,22]. In concrete, as well as in a cement matrix, the major component is cement. When testing concrete samples, fluctuations in reference samples from other batches may be characterized by significant differences in strength, which may ultimately affect how individual results relate to each other. The authors of the article dealt with the problem of assessing the failure of fibre cement boards by reviewing the scientific literature and drawing conclusions from it and then conducting experimental research. The experimental tests were performed on samples taken from the actual model of the elevation that had been exposed to fire. Samples were taken from various locations and then analysed and assessed. The assessment was made on the basis of the analysis of AE signals recorded during the course of three-point bending, and the assessment of strength during three-point bending. On the basis of these analyses, the state of destruction was determined, and conclusions were drawn.

## 3. Research Model and Testing Method

The actual façade model was a ventilated façade attached to a test platform, which was also the substrate on which the fire impact and progression tests were performed. Large-scale models are a very good way to study fire development and evaluate individual façade elements in terms of flammability, the behaviour of individual materials during a fire and the fire safety of the whole system. Large-scale studies may produce different results and present different critical areas with respect to studies on small samples.

The façade cladding analysed in this paper, installed on the real model, was made of 8 mm-thick fibre cement boards of natural colour, i.e., not dyed in the mass. It was mounted with mechanical connections to an aluminium substructure. The substructure consisted of metal vertical profiles mounted mechanically via brackets to the ground. There was a fire coming out of the combustion chamber, which was affecting the façade with high temperatures, realising the scenario of a fire in the room spreading through the window opening to the façade. A ventilation device was placed in the combustion chamber on the rear wall, which allowed the reproduction of a real fire situation. The division of the façade into cladding panels and the location of thermocouples and sampling sites are shown in Figure 3. The dimensions of the actual large-scale façade model were ~3 m × 3.5 m, and it was attached to a wall made of autoclaved aerated concrete, 600 kg/m^3^ variety, with dimensions of 3.98 m × 3.98 m.

Figure 4 presents the large-scale study model during the course of the study along with the actual sampling location.

Subsequently, samples D5, D4 and D3 from the large-scale fire-affected model were taken from the above combustion chambers; they fell off at 13:30 min, 17:15 min and 34:00 min, respectively. The temperature course was recorded by the thermocouples TE3, TE7 and TE9 for the samples D5, D4 and D3, respectively. In the case of sample D5 located in the immediate area of the fire source, the temperature reached over 600 °C. Conversely, for sample D4, the temperatures were between 400 and 500 °C, and for sample D3, between 300 and 400 °C. The dimensions of the samples taken were 20 mm × 100 mm, with a thickness of 8 mm.

In order to identify the effect of high-temperature exposure on the degradation of fibre cement boards, tests were carried out using the acoustic emission method during a three-point bending test. Similar issues on fibre cement boards have also been analysed by other authors [23,24]. The test samples were cut from the façade board subjected to the fire according to their location shown in Figure 4. The cut-out samples were 20 mm × 100 mm in size. They were then placed in a testing machine, and a three-point bending test was carried out. For the three-point bending test, the PASCAL MIKROPRASA P—3 kN was used, with a load range between 0 and 3 kN. A constant crossbeam travel increment of 0.1mm/min was set during the testing of the fibre cement samples. The three-point bending test stand with the acoustic emission measurement setup is shown in Figure 5.

The analysis of the obtained results of three-point bending tests took into account the course of bending stresses *σ_m_*, the modulus of rupture (*MOR*), the number of EA events *N_zd_* and the sum of EA events *∑N_zd_*. The modulus of rupture (*MOR*) was determined from the standard formula [5]:(1)$$MOR=\frac{3Fl_{s}\;}{2b\;e^{2}}$$
where:*MOR*—modulus of rupture [MPa],*F*—load (force) [N],*l_s_*—length of the support span [mm],*b*—sample width [mm],*e*—sample thickness [mm].

In addition, in order to verify the degraded element samples taken after testing, a structure analysis was performed on a Keyence VHX-7000 series microscope. The digital microscope used to verify the structure of the fibre cement boards and the test stand are shown in Figure 6.

The device was equipped with a digital microscope unit which, depending on the type of lens, allowed taking pictures within the zoom range of ×20–×4000 (a wide-angle lens was used in this study, as well as a standard lens with a maximum zoom of ×200).

## 4. Study Results

The evaluation of the destruction of fibre-cement board structures under the influence of fire was made through the analysis of EA signals recorded during the course of the three-point bending test and was based on EA descriptors such as the number of events *N_zd_* and the sum of events *∑N_zd_*. Figure 7, Figure 8, Figure 9 and Figure 10 show a recording of the event rate N_zd_ and bending stress *σ**_m_* versus time for selected samples designated D3, D4, D5 and, for comparison, an untreated reference board. Table 1 summarises the resulting modulus of rupture (*MOR*) values and *∑N_zd_* event sum values for the tested samples.

When analysing Figure 7, Figure 8, Figure 9 and Figure 10, it is clear that the fire had a clear effect on the values of the EA descriptors recorded in the three-point bending test of the fibre cement boards in question. Thus, the recording of the rate of *N_zd_* events as a function of time for the reference board (Figure 7) differed significantly from that obtained for the fire-treated boards. In the case of the reference board, it can be seen that from about the 140th second of the test, the rate of *N_zd_* events started to increase steadily until the maximum value of the *MOR* modulus of rupture was reached, at which point there was a clear spike in this descriptor. In the case of the samples labelled D3, D4 and D5 (Figure 8, Figure 9 and Figure 10, respectively), one can see a completely different course of degradation. Namely, there was a sharp spike (peak) in the recorded EA event rate values around the time the test samples reached the modulus of rupture (*MOR*). Analysing the results presented in Table 1, it can be seen that a significantly higher sum of EA events was recorded for the reference sample. In contrast, for samples D3, D4 and D5, a clear decrease in the recorded sum of EA events can be seen. A trend can be observed: as the temperature to which the tested samples were exposed and the time of exposure increased, the sum of EA events decreased. In analysing the values of the obtained modulus of rupture (*MOR*), a similar trend was noticed. The reference plate had the highest modulus of rupture (*MOR*) (43.45 MPa). With the increase of the temperature to which the boards were subjected during the fire, there was a clear decrease in the modulus of rupture (*MOR*), ranging from a value of 14.83 MPa for board D4 through to 1.72 MPa for board D3, with a value of 0.22 MPa for board D5.

The above results indicate that as the temperature and the time of its action on the subject boards during the fire increased, the nature of the sample degradation course during the three-point bending test changed significantly, as can be seen from the recorded acoustic emission descriptors and the obtained modulus of rupture s (*MOR*). It can be concluded unequivocally that a high temperature has a destructive effect on the structure of fibre cement boards. Thus, analysing the results obtained using the acoustic emission method shown in Figure 7, Figure 8, Figure 9 and Figure 10, it can be seen that the boards subjected to fire were damaged by brittle fracture when the modulus of rupture (*MOR*) was reached. This was evidenced by the EA event rate recording, which was essentially a single peak compared to what observed for the reference board. On this basis, the authors conclude that a high temperature caused damage to the cellulose fibres (pyrolysis) contained in the board structure. The brittle fracture recorded during the tests may be indicative of the complete pyrolysis of the cellulose fibres and, therefore, of the resulting structure of the board, consisting only of the cement matrix. This was particularly evident in samples D3 and D5.

In order to verify the destruction of the structure of the boards subjected to fire, the authors additionally performed an analysis of the fracture structure using a digital microscope. Figure 11 shows a view of the fracture of the reference sample after the three-point bending test. It should be noted that the fractured sample showed fibres that were destroyed by pulling or tearing, as well as delamination—the result of manufacturing technology.

Figure 12 shows a view of the sample D5 fracture after the three-point bending test. A change in the structure of the board could be seen through the precipitation of individual cement matrix materials (known as crystallisation) and the delamination of the fibre cement board. Delamination was caused by the temperature gradient between the outer surface and the inner surface (mounted from the platform) and the total thermal expansion of the material. In addition, complete pyrolysis of the fibres could be seen—caverns and voids left after the melting of the fibres (Figure 12b). The fracture shape was straight, which indicated that it followed the shortest line. From this figure, it is clear that the sample was completely destroyed.

## 5. Discussion

All samples were taken from the actual façade model, from materials that fell off during the test. It should be noted that the samples did not show a trend for the measured values, the lowest modulus of rupture (*MOR*) being that of sample D5, which was located closest to the combustion chamber. The higher modulus of rupture (*MOR*) was found FOR sample D3, which was located 120 cm above the combustion chamber. In contrast, sample D4 located 80 cm above the combustion chamber showed a much higher strength (*MOR*) than samples D3 and D5. It should be noted that for all locations sampled, the fibre melting point and the temperature at which the cement matrix degraded were exceeded, which, for the area directly above the fire source, amounted to a maximum of 28% of the number of EA events of the reference sample. The loss of the modulus of rupture (*MOR*) is also worth noting. For the degraded sample D4, this was up to 34% (14.83 MPa) of the strength of the reference sample (43.45 MPa).

When we examined sample degradation by the action of fire temperatures, we observed a completely different course of degradation of the tested boards with respect to the reference board. Namely, there was a sharp spike (peak) in the recorded EA event rate values around the time the test samples reached the modulus of rupture (*MOR*). This indicates that destruction occurred by brittle high-energy fracture, with a low number of EA events. This was also found in other studies. Szymków, in his work [18], also showed a similar convergence—i.e., degraded samples were more brittle, were characterized by the lack of the “flow” phase, showed brittle high-energy cracking with a small number of EA events.

The conducted tests showed that a significantly higher sum of EA events was recorded for the reference sample. In contrast, a clear decrease in the recorded sum of EA events was seen for the fire-treated samples. The following trend was observed: as the temperature to which the tested samples were exposed and the time of exposure increased, the sum of EA events decreased.

The analysis of the obtained modulus of rupture (*MOR*) values showed a similar trend. The reference plate had the highest modulus of rupture (*MOR*) (43.45 MPa). With the increase of the temperature to which the boards were subjected during the fire, there was a clear decrease in the modulus of rupture (*MOR*), ranging from a value of 14.83 Mpa for board D4 through to 1.72 Mpa for board D3, with a value of 0.22 Mpa for board D5.

Digital microscope studies of the degraded samples indicated complete pyrolysis of the fibres and crystallisation of the internal composite structure. We also observed significant delamination of the sample due to the temperature gradient acting on it, i.e., the outer and inner layers of the cladding surface, as well as a significant value of the expansion of the sample (caused by high temperatures).

## 6. Summary

The effects of high temperatures are inherently destructive to most building products. Heat resistance is defined as the period of time during which a product retains the desired properties. It is worth noting that large-scale studies provide very significant opportunities for the analysis of façade cladding exposed to high temperatures caused by fire. Fire temperatures occurring directly above the fire source are critical for this composite and reach values between 500 °C and 700 °C. In order to determine the destructive effect of high temperature on the structure of the fibre cement boards tested, samples were examined in a three-point bending test using the acoustic emission method. In addition, in order to analyse further the effect of fire on the structure of the boards, board samples were observed using a digital microscope. The fibre cement boards tested in this study showed different ranges of resistance to fire. The most important observations and conclusions formulated by the authors on the basis of the obtained results are summarised below.

In the destruction critical zone, i.e., where the fibre melting point and the temperature at which the cement matrix is destroyed have been exceeded, the samples cannot be used repeatedly.Critically degraded samples are destroyed by a sharp spike (peak) in the recorded EA event rates.Research using a digital microscope is sufficient to assess the complete degradation of the fibre cement composite materials, in terms of the assessment of fibre pyrolysis and the crystallization of the internal structure of the composite

Summarising the information presented above, the authors would like to point out that the tests carried out in this study are very important from the point of view of construction practice, particularly due to the fact that little information is available in the literature on the behaviour of ventilated façades and, thus, of the cladding installed on them at high temperatures, i.e., under fire conditions. Furthermore, it should be emphasised that the above tests are only pilot tests. The authors are investigating fire-exposed fibre cement boards through other methods. The results of these studies will be the subject of further scientific work.

## Figures and Tables

**Figure 1 materials-15-02929-f001:**
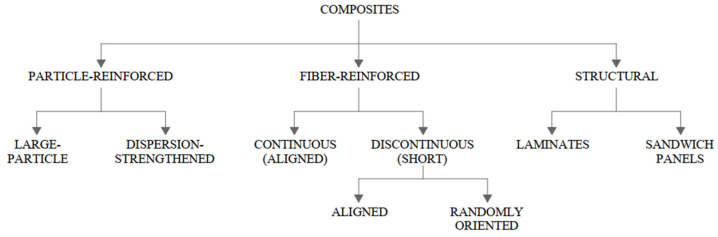
Classification of composite materials.

**Figure 2 materials-15-02929-f002:**
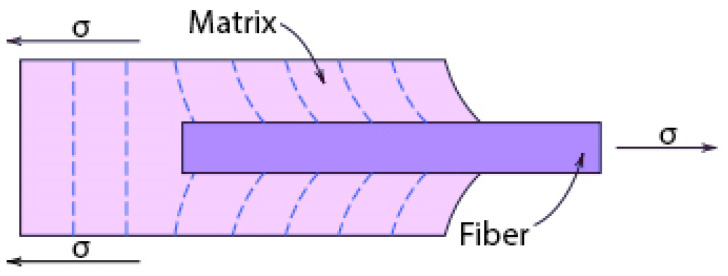
Deformation pattern in the matrix surrounding a fibre that is subjected to tension.

**Figure 3 materials-15-02929-f003:**
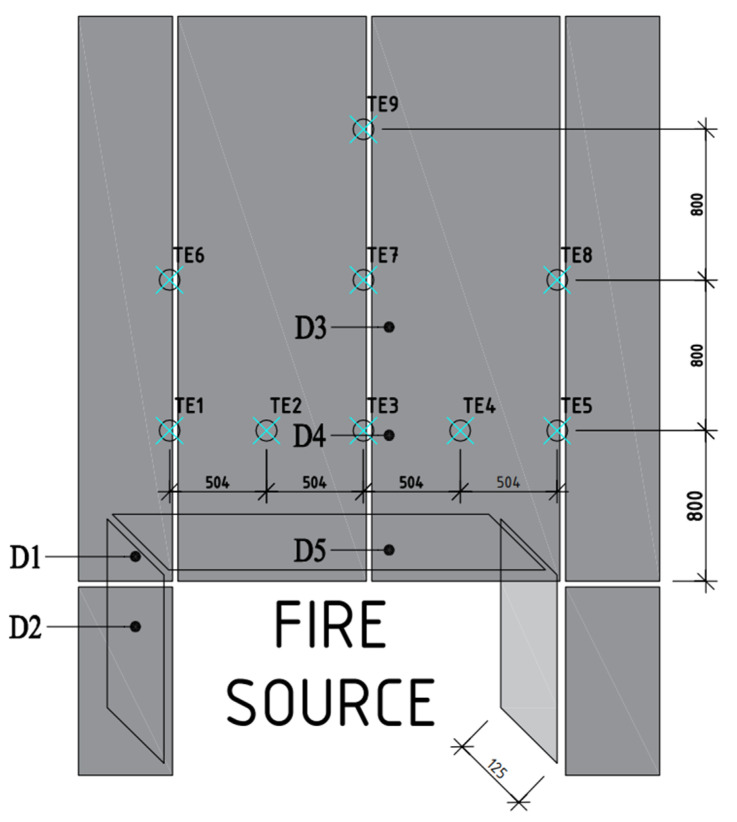
Layout of the exterior cladding panels and thermocouples along with sampling location.

**Figure 4 materials-15-02929-f004:**
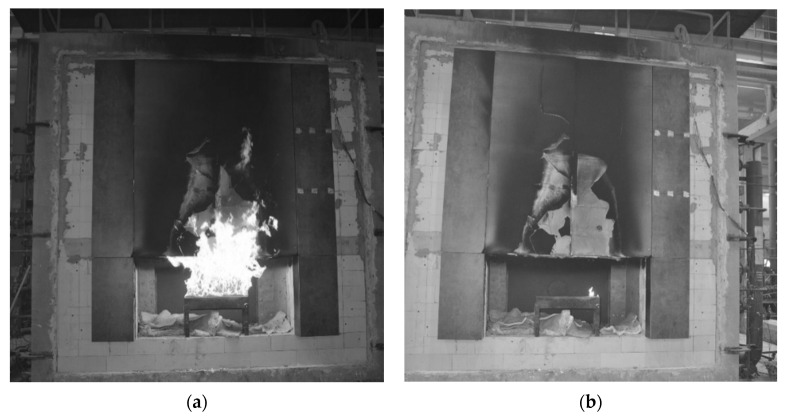
Large-scale model: (**a**) during the study; (**b**) after the study, with the indication of the sampling location.

**Figure 5 materials-15-02929-f005:**
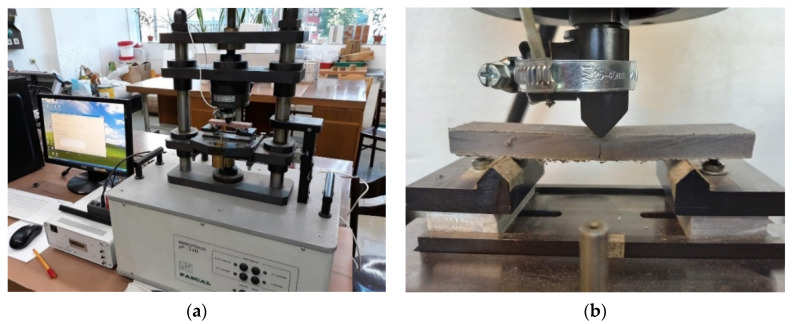
Test stand for acoustic emission measurements and the fibre cement board used during the test: (**a**) view of the testing machine; (**b**) view of the specimen during its destruction.

**Figure 6 materials-15-02929-f006:**
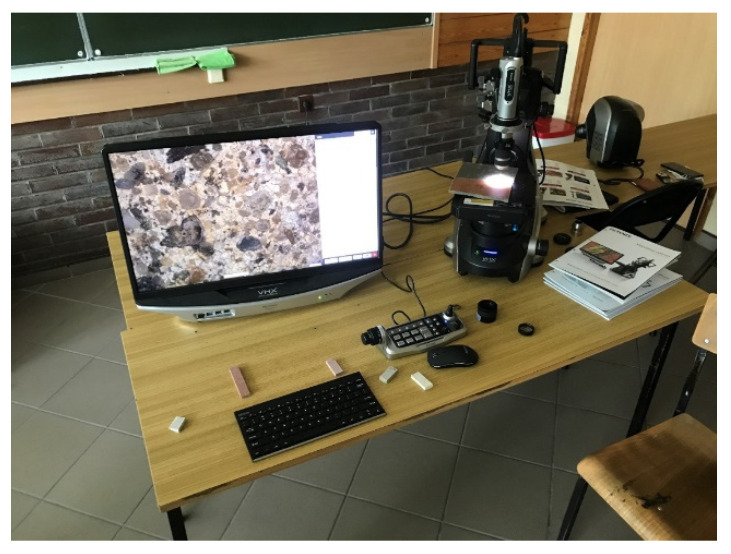
Keyence VH×-7000 series digital microscope.

**Figure 7 materials-15-02929-f007:**
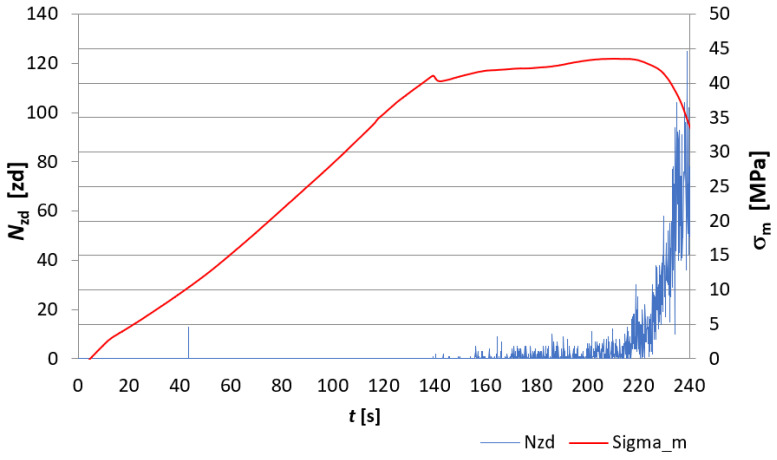
Recording of the event rate *N_zd_* and bending stress *σ**_m_* versus time for the reference board.

**Figure 8 materials-15-02929-f008:**
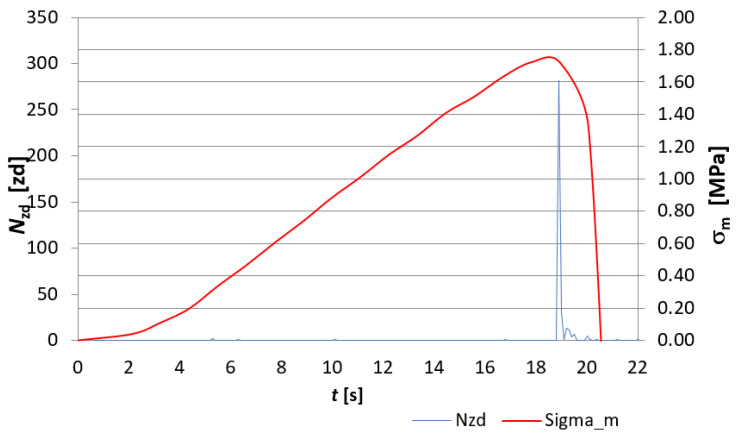
Recording of the event rate *N_zd_* and bending stress *σ**_m_* versus time for the D3 board.

**Figure 9 materials-15-02929-f009:**
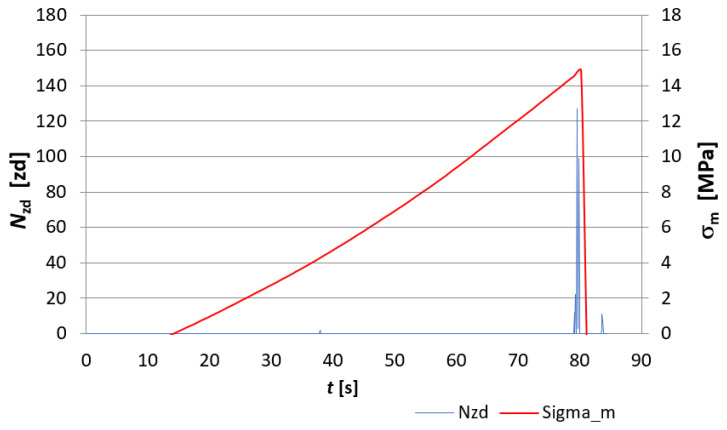
Recording of the event rate *N_zd_* and bending stress *σ_m_* versus time for the D4 board.

**Figure 10 materials-15-02929-f010:**
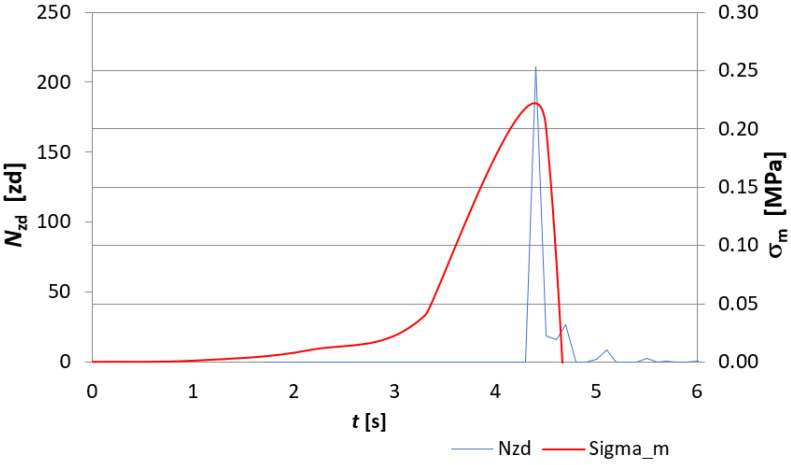
Recording of the event rate *N_zd_* and bending stress *σ_m_* versus time for the D5 board.

**Figure 11 materials-15-02929-f011:**
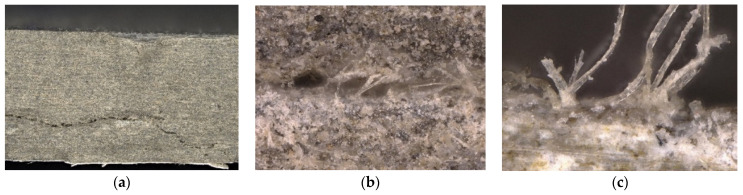
View of the reference sample under a digital microscope: (**a**) magnification ×20, (**b**) magnification ×150, (**c**) magnification ×400.

**Figure 12 materials-15-02929-f012:**
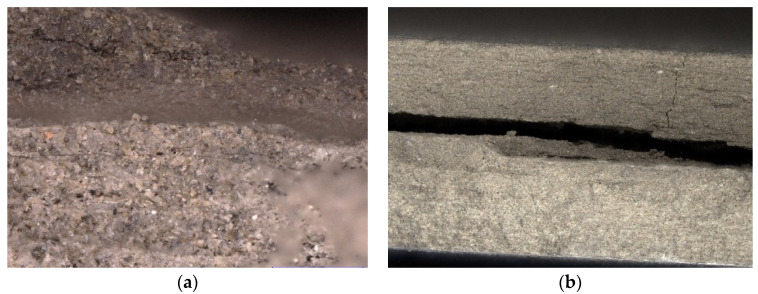
Example view of the D5 sample under a digital microscope: (**a**) magnification ×150, (**b**) magnification ×20.

**Table 1 materials-15-02929-t001:** Summary of averaged modulus of rupture (*MOR*) and example event sum *∑N_zd_*.

Sample Identification	Bending Strength (*MOR*) [MPa]	Sum of Events *∑**N_zd_* [–]
Reference	43.45	1715
D3	1.72	228
D4	14.83	475
D5	0.22	198

## Data Availability

The data presented in this study are available on request from the corresponding author.

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
