# Peer review of "Assessment of the Destruction of a Fibre Cement Board Subjected to Fire in a Large-Scale Study"

_materials, 2022, doi:10.3390/ma15082929_

Round 1
Reviewer 1 Report
This article presents the results of a study involving the assessment of the structural destruction of fiber cement boards taken from a large-scale model subjected to fire. These were then subjected to a three-point bending test using the acoustic emission method.
The topic is significant in material engineering. The manuscript has some of its technical merits. The structure of the manuscript is also well. It should fall into materials Journal. However, several questions might pay attention to:
- There are some typos in the manuscript. Recommend the authors should check it carefully before submitting it.
- The authors introduced the Fiber-cement boards. It is well known to the readers. The authors also provided some citations. Figures 1 and 2 could be removed from the text.
- The authors should briefly introduce the whole paper at the end of section 1. It will be better for the reader to follow up.
- The image size is large, and the label font size is also large in Figure 4. The authors should set it in a suitable size.
- The authors ignored the sub-caption of two images in Figure 6. Please add it.
- The sub-caption of Figure13 is in upper Figure 13 and is not in the correct location. Please check it.
- There is no discussion section in the text. The authors have not compared previous literature. Recommend authors should extend your discussion.
- The conclusions include many detailed results. It should move to section 3. Recommend the authors explore the analysis results briefly and give several valuable points for engineering in section 4. The authors could also give some future expectations in the section.
Hopefully, this will help in the revision of the manuscript.
Reviewer 2 Report
This manuscript lacks in-depth analysis. Just listing observation from experiments are not enough for a journal paper. Also, innovation of the presented work is very limited. I do not see any improvement compared to existing works. Compare to the large-scale model mentioned here, previous research with small-scale models but various conditions can also give similar results. Also, there are some other minor issues in the writing:
- When you wrote stresses, it is always like "stresses m", like on row 16. Is this a typo?
- What's the thickness of the cut-out samples on row 181?
- Temperature profiles measured by thermocouples should be provided at least for the locations near the cut-out samples, as only in this way the "temperature to which the tested samples are exposed to" and "the time of exposure" mentioned on rows 244 and 245 can be quantified.
Reviewer 3 Report
- The article still needs several grammatical and syntax improvements. Use of English service center is recommended.
- The abstract is written qualitatively. Majority of the qualitative statements should be modified for quantified result comparisons.
- The introduction needs to be revised for higher quality language. The authors mentioned some works without stating about the contributions, pros and cons and the how the current work would address. The language should be improved.
- The authors are required to add a paragraph of purpose describing the various general procedures to address harsh environmental conditions. The following references should be added for this statement.
- Experimental investigation of sound transmission loss in concrete containing recycled rubber crumbs.
- Compressive behavior of concrete under environmental effects. IntechOpen.
- Temperature and humidity effects on behavior of grouts. Advances in concrete construction, 5(6), 659.
- Nano silica and metakaolin effects on the behavior of concrete containing rubber crumbs. CivilEng, 1(3), 264-274.
- In addition for the statement “The fibrecement boards manufactured nowadays consist of cement, cellulose fibres, synthetic fibres, and various additives and admixtures, i.e. limestone dust, mica, perlite, kaolin, microsphere ’ The following references should be added for this statement
- Investigation of steel fiber effects on concrete abrasion resistance, Advances in concrete construction, 9(4), 367-374.
- More structural details to Figure 4 and Figure 6 is required.
- Figure 7 is unnecessary and should be substituted with more detailed information.
- Figure are not high quality, if they are taken from other resources, they have to be re developed and properly referenced. Also, the legend and titles should be consistent. Figs 8-11 should be professionally made, and legends and titles should be clear and not crossing.
- What temperature causes Delamination and what is the treason?
Round 2
Reviewer 1 Report
The manuscript has been revised according to the most of points.
Author Response
I am enclosing herewith a manuscript entitled: “ Assessment of destruction of a fibre-cement board subjected to fire in a large-scale study” submitted to “Materials” after corrections with detailing (highlighted by yellow color) any changes and replay to Reviewers on separate pages. The linguistic errors have been corrected. The paper has been checked by a sworn translator of the English language.
Reviewer 2 Report
The authors have revised the manuscript accordingly. With addition of more discussion, scientific soundness of this work is presented clearly this time.
Author Response

(The authors gave the same response as above.)

Reviewer 3 Report
The English language could be improved.
Author Response

(The authors gave the same response as above.)
